# Lattice Boltzmann Modeling of Additive Manufacturing of Functionally Graded Materials

**DOI:** 10.3390/e27010020

**Published:** 2024-12-30

**Authors:** Dmytro Svyetlichnyy

**Affiliations:** AGH University of Krakow, Faculty of Metals Engineering and Industrial Computer Science, al. Mickiewicza 30, 30-059 Kraków, Poland; svyetlic@agh.edu.pl; Tel.: +48-692-983-805

**Keywords:** functional graded materials, powder bed fusion, lattice Boltzmann method, modeling, simulation, selective laser melting

## Abstract

Functionally graded materials (FGMs) show continuous variations in properties and offer unique multifunctional capabilities. This study presents a simulation of the powder bed fusion (PBF) process for FGM fabrication using a combination of Unity-based deposition and lattice Boltzmann method (LBM)-based process models. The study introduces a diffusion model that allows for the simulation of material mixtures, in particular AISI 316L austenitic steel and 18Ni maraging 300 martensitic steel. The Unity-based model simulates particle deposition with controlled distribution, incorporating variations in particle size, friction coefficient, and chamber wall rotation angles. The LBM model that simulated free-surface fluid flow, heat flow, melting, and solidification during the PBF process was extended with diffusion models for mixture fraction and concentration-dependent properties. Comparison of the results obtained in simulation with the experimental data shows that they are consistent. Future research may be connected with further verification and validation of the model, by modeling different materials. The presented model can be used for the simulation, study, modeling, and optimization of the production of functionally graded materials in PBF processes.

## 1. Introduction

Gradient materials (functionally graded materials, FGMs) are materials in which a continuous change in functional or construction properties has been achieved along at least one specific direction in a selected technological process. These materials have been known for many centuries, although their current name only appeared recently. FGMs allow for unique properties and behavior, tailored properties and multifunctional capabilities, thermal resistance, corrosion and wear resistance, improved mechanical performance, light weight, and high strength, and are widely used in aerospace, automotive, biomedical, energy, and other applications. Details of the concept of functionally graded materials, their use and fabrication processes can be found in [1]. Different aspects of FGM, state-of-the-art research, and development findings can also be found in [2].

Naebe and Shirvanimoghaddam [3] presented an overview of advances in FGM research. Materials selection, and the fabrication, characterization, analysis, and modeling of FGMs is the focus of the overview. Additionally, challenges in the fabrication of FGMs are discussed. Another overview of Kieback et al. [4] presents the achievements in the field of processing techniques involving metal melting and sintering. They considered modeling the formation of gradients in the microstructure and presented examples of numerical simulations. DebRoy et al. [5] reviewed the process, structure, and properties of additive manufacturing (AM) of metallic components. They described AM processes that consolidate materials such as powder, wire, or sheets into a dense metallic part by melting and solidifying with the aid of an energy source such as a laser, electron beam, or electric arc, or by the use of ultrasonic vibration in a layer-by-layer manner. One of the recent reviews [6] is a good guide on the current state of application areas, fabrication possibilities, and challenges in the field of FGMs. Yadav et al. [6] presented seven AM techniques that are used most frequently for the fabrication of FGM materials: vat photo polymerization, material extrusion, material jetting, sheet lamination, binder jetting, powder bed fusion (PBF), and direct energy deposition.

PBF is one of the most common AM technologies used to manufacture FGMs. PBF includes selective laser melting (SLM) and selective laser melting (SLS) processes, which allow for precise control of the composition and microstructure of FGMs. Mussatto [7] presented a review of several techniques of material deposition in the multi-material PFB process: conventional spreading, patterning drums, spreading plus suction, vibrating nozzle, hopper feeding, alternating, and electrophotographic.

Tian et al. [8] consider metallic FGMs and their processes. They noted that metallic FGMs can be classified in different ways: structural (porosity, grain size) and compositional, coating and bulk, and continuous and discontinuous. They also described 3D printing or AM techniques including PBF and experimental results on chemical composition and microstructure. Wu et al. [9] presented a review of the experimentally observed mechanical and microstructural characteristics of interfaces in multi-material laser PBF. They described mechanical spreading, nozzle-based, electrophotographic, and hybrid techniques that make it possible to achieve compositional gradients and discrete boundaries in two and three dimensions. They also show the challenges related to material transitions, such as defects, segregation, and phase separation.

Bandyopadhyay and Traxel [10] highlighted fundamental modeling strategies, considerations, results, and validation techniques using experimental data for application-optimized designs for metal additive manufacturing. They discussed the potential for modeling and how simulation would enhance the metal-AM optimization process.

Recent reviews on laser PBF modeling and simulations can be found in [11,12,13]. Multi-material AM processes modeled by Shinjo and Panwisawas [14] used the coupled level-set and volume-of-fluid method. They modeled the formation of the melt pool, the keyhole, and the flow for different elements. Küng et al. [15] presented a multi-material model for the lattice Boltzmann-based simulation of the PBF process of a binary alloy as a mixture of two elements. The heat capacities, latent heat, and other parameters are considered to be concentration-dependent. Melting and solidification were modeled according to the phase diagram. They simulated a single melt track in a multi-material powder mixture. Tang et al. [16] presented simulations of the multi-material laser PBF process using a volume-of-fluid model. They modeled two- and three-component powders with mixed titanium Ti, niobium Nb, vanadium V, and arsenic Ar particles.

Wang et al. [17] presented a coupled finite volume-discrete element model for SLM on powder scale. They modeled a multilayer, multitrack process. Fotovvati and Chou [18] presented a model based on the discrete element method and computational fluid dynamics for the simulation of the multilayer PBF process, creating 10 layers.

Recently, the author participated in the creation and development of the SLM and SLS processes platform, which allows for multistage multi-material simulations [19,20,21]. However, the scheme used in the previous simulations is based on the manufacturing process using metal alloys and bioactive glass alternately that cannot be applied for simulation of the PBF process of FGM fabrication.

It should be noted that it is very difficult to find publications with modeling of the process for FGMs. Among the various methods of material deposition that can be found in [7], the method proposed and presented in [22,23] was chosen for the simulation.

The objective of this paper is to present a modified PBF model that allows for the simulation of fabrication of the functionally graded material. To achieve this goal, the Unity-based model was adopted for the chosen deposition method, and the LBM-based model was extended with the diffusion model and concentration-dependent thermal and mechanical properties. The results of the simulation and comparison with the experimental data are also presented in the paper.

## 2. Model of PBF Processes

### 2.1. Holistic Model and Main Algorithm

The model of PBF processes is presented in Figure 1. First, the LBM-based model was developed for SLM. Details of the development, testing, verification, and validation of the SLM model and its submodels can be found in previous papers [19,20,21,24,25]. A platform was created for the 3D simulation of additive layer manufacturing processes characterized by changes in the state of matter (melting–solidification) [20]. Integration of the Unity-based powder bed generation model into the platform allowed for multistage multilayer multi-material simulations [19]. Then, it was shown that LBM allows for modeling not only of the SLM but also of the SLS process [21].

The model presented in this paper is further developed in the direction of a simulation of a mixture of two materials. The previous multi-material model allowed for the simulation of two materials in a sequential way. The first material can be processed in several cycles with one or more layers. After that, the rest of the powder was removed and the powder of the second material was deposed. The second material can also be processed in several cycles with one or more layers. The first material (metals or alloys) with a higher melting temperature can be used as a matrix, and the second material (glasses or ceramics) can be used as a filler or coverage. The materials were not mixed.

The model is now extended with an additional diffusion model, which allows for the modeling of mixtures of different materials. Two materials with unlimited solubility to each other are chosen to simplify the model. They are AISI 316L austenitic steel and 18Ni maraging 300 martensitic steel. These materials can be used for the fabrication of functionally graded materials according to the scheme shown in [22,23].

The model contains the submodels presented in Figure 1, which are associated with the physical processes and phenomena that accompany the technological processes of PBF. Furthermore, the holistic model presented in [19,20] was supplemented with the possibility of modeling a mixture of two materials at the same time and diffusion. The powder removal remains in the model but is not used in calculations with mixed materials. The six basic processes and five submodels are considered at the second and third levels. Most of them are described in previous publications [19,20,21,24,25]; some elements are described in the following sections.

The main algorithm represented as a block diagram (Figure 2) was also modified compared to the previous algorithm [19,20]. The algorithm has two circuits. The first external circuit contains the powder bed generation (Unity-based model), cycle initialization, cycle modeling (LBM calculation module), and powder removal, which is not used here.

The second internal circuit of the algorithm (shown in blue inside the ‘LBM calculation module’) performs one cycle of the PBF process. Operations are multiple repeated according to the presented sequence, which contains the laser beam movement, heating, and calculation of the new temperature, calculation of normal to the liquid and solid surfaces, calculation of the fluid density and velocity, collision for fluid, heat flow and diffusion, streaming for fluid, heat flow and diffusion, boundary condition, and movement of the liquid–gas interface.

The external cycle prepares the simulation of one layer, while the internal one realizes this simulation. The internal cycle is multiple repeated modeling one or many tracks, calculating small steps of laser movement and pauses after each track. The external cycle can be repeated as many times as many layers are modeled. In this paper, only a single track of the single layer is presented. The submodels, subroutines, and kernels of this circuit are essential for the proper functioning of the entire platform. Some of these models are described below; the others can be found elsewhere [19,20,21].

### 2.2. Powder Bed Generation Model

The Unity-based model was presented in [19,21]. This model was created by using the publicly available Unity game engine (https://unity.com, accessed on 20 December 2024). This model is modified to simulate the deposition of the powder of two materials. The modeling scene created in Unity is presented in Figure 3a. The primary elements of the scene are a powder chamber with parts separated by a thin wall with different powders, building platform, and coater. The Unity coordinates set is also presented. The thin wall can rotate around the vertical axes (axes Y) located in the center of the powder chamber (supply platform). A rotation angle is set before filling parts of the chamber with powder of different materials. The floor of the chamber can move vertically at the chosen level. The coater can then move horizontally along the Z axes, distributing the powder on the building platform. After that, the coater returns to the initial location. Finally, the floor of the building planform goes downward to the height of one layer, and the cycle with the movement of the floor of the powder chamber, the blade of the coater, and the floor of the building platform can be repeated. Two cases of filling of the powder chamber are also presented in Figure 3. The thin wall separating the parts of the chamber can either be rotated or not (Figure 3b). In the second case, this wall is rotated at an angle of 25° (Figure 3c). After rotation, the parts are filled with the appropriate powder material. The sizes of the powder chamber and the building platform are the same and equal to 25 × 25 mm in the Unity model, which represents the real domain of 1250 × 1250 μm. This means that the Unity model is created on a scale of 20:1 to respond to the real domain.

### 2.3. LBM Models of the PBF Process

LBM was developed from lattice gas automata [26] and is derived from the Boltzmann equation, a fundamental equation in statistical mechanics that can be considered as one of the computational fluid dynamics (CFD) methods. Several differential equations can be reduced to or approximated by the Lattice Boltzmann equation (LBE): Navier–Stokes, advection–diffusion, energy (heat conduction), Poisson, shallow water, elastic wave, Burgers, and magnetohydrodynamic. Details on how to approximate these equations can be found elsewhere [27,28].

LBM simulates the microscopic dynamics of particle distribution functions on a discrete lattice grid. These particle interactions and streaming processes yield macroscopic properties like density, velocity, temperature, enthalpy, concentration, etc. LBM uses discrete time, space, and velocity and calculates macroscopic properties through collision and streaming operations [27,28,29].

The approximations of different partial differential equations by LBE open up wide possibilities for multiphysics coupling and modeling. The algorithm of LBM is easy to implement due to the explicit streaming and collision steps and is highly suited for parallel computations on CPU and GPU. LBM operates effectively with complex boundaries. Thus, the model of the PFD process takes into account fluid flow with free surface (flow, surface tension, wettability, convection, etc.), heat flow (heat source, heat transfer, heat conduction, melting and solidification), and diffusion.

The LBM model of the PBF process is based on the previously developed model for the modeling platform, and most of the submodels can be found in detail in previous publications [19,20,21,24,25]. They contained two cooperative parts that are responsible for the fluid and heat flow processes. Fluid flow was considered for a single material.

In this paper, the flow of the mixture is simulated. To determine the composition of the mixture, a volume fraction of one material in the mixture is introduced. Fluid flow does not change the composition of the mixture. The composition of the mixture is changed by diffusion. Diffusion is simplified. Instead of considering the diffusion of many elements in different mixtures, a mixture is considered as a single homogeneous material in which both materials are diffused at the same rate, defined by a common diffusion coefficient.

The common algorithm of the LBM model is presented in Figure 2 as the internal circuit. Each of the three problems (fluid and heat flow and diffusion) is solved according to the same scheme: calculating macroscopic values based on the input distribution, determining the equilibrium distribution based on macroscopic values, determining the output distribution in the collision operation, and determining the input distribution based on the output distribution in the streaming operation by transferring the components of the distribution function to appropriate neighboring nodes, taking into account boundary conditions and other factors. Since all the mechanical and thermal properties of a material depend on chemical composition and temperature, all three problems are interdependent and can be solved simultaneously as much as possible. This explains the calculation sequence, which is described in detail in Section 2.3.4.

Such blocks as the calculation of macroscopic variables, collision and streaming operations for fluid and heat flow and diffusion are presented in the following Section 2.3.1, Section 2.3.2 and Section 2.3.3; the other blocks can be found in previous publications [19,20,21,24,25].

The D3Q19 velocity model is applied for all the tasks. Therefore, 19 components of the distribution functions that represent directions and velocities are used in the 3D models.

#### 2.3.1. LBM Model of Fluid Flow

Macroscopic variables, that is, the density *ρ* of the mixture fluid and the velocity **v**, are calculated according to the following equation (vectors are indicated in bold):(1)$$\rho =\sum_{i=1}^{19}f_{i}^{in}$$
(2)$$\rho v=\sum_{i=1}^{19}f_{i}^{in}e_{i}$$
where *f_i_*^in^—appropriate component of input distribution function of the liquid flow model, **e**—phase space variable—velocity (vector), set of discrete velocities, **v**—velocity vector {*vx*, *vy*, *v_z_*}, and *i*—component of the velocity model.

The collision operation for fluid flow calculates the output distribution function ***f***^out^ based on the input ***f***^in^ and equilibrium ***f***^eq^ distribution functions. Each distribution function contains 19 components. Then, the equilibrium ***f***^eq^ and output distribution functions are defined as follows:(3)$$f_{i}^{eq}=w_{i}\rho \left[1+3e_{i}\cdot v+4.5{\left(e_{i}\cdot v\right)}^{2}-1.5v\cdot v\right]$$
(4)$$f_{i}^{out}=f_{i}^{in}+\frac{∆t}{\tau_{f}}\left(f_{i}^{eq}-f_{i}^{in}\right)+F_{i}$$
where *w_i_*—weights, *F*—external force, for example, gravity, Δ*t* = 1—time step, and *τ_f_*—relaxation time for fluid flow.

The streaming operation that defines input distribution function is performed according to:(5)$$f_{i}^{in}\left(x+e_{i},\;t+1\right)=f_{i}^{out}\left(x,\;t\right)$$
where **x**—node coordinates.

#### 2.3.2. LBM Model of Heat Flow

The new temperature *T* is calculated as follows:(6)$$T=Q+\sum_{i=1}^{19}g_{i}^{in}$$
where *g*^in^—distribution function of the heat flow model, *Q*—heat source.

The collision operation is carried out:(7)$$g_{i}^{eq}=w_{i}T\left[1+3e_{i}\cdot v+4.5{\left(e_{i}\cdot v\right)}^{2}-1.5v\cdot v\right]$$
(8)$$g_{i}^{out}=g_{i}^{in}-\frac{1}{\tau_{h}}\left(g_{i}^{in}-g_{i}^{eq}\right)$$
where *τ_h_*—relaxation time for heat flow.

And finally, the streaming operation is as follows:(9)$$g_{i}^{in}\left(x+e_{i},\;t+1\right)=g_{i}^{out}\left(x,\;t\right)$$

#### 2.3.3. LBM Diffusion Model

This is a new part of the model. The fraction of the first material in the mixture (or its concentration) *C* is calculated as follows:(10)$$C=\sum_{i=1}^{19}c_{i}^{in}$$
where *c*—distribution function of the concentration flow.

The collision operation is as follows:(11)$$c_{i}^{eq}=w_{i}C\left[1+3e_{i}\cdot v+4.5{\left(e_{i}\cdot v\right)}^{2}-1.5v\cdot v\right]$$
(12)$$c_{i}^{out}=c_{i}^{in}-\frac{1}{\tau_{d}}\left(c_{i}^{in}-c_{i}^{eq}\right)$$
where *τ_d_*—relaxation time for diffusion.

The streaming operation for diffusion is calculated as follows:(13)$$c_{i}^{in}\left(x+e_{i},\;t+1\right)=c_{i}^{out}\left(x,\;t\right)$$

#### 2.3.4. LBM Cycle and Model Interaction

The second internal circuit of the algorithm (Figure 2) begins with calculations of macroscopic variables such as temperature (6) and material fraction (10). However, the temperature model is more complex than (6); it takes into account melting and solidification, and its details are described in a previous publication [20]. Here, all the properties of the material that depend on temperature and material fraction are also calculated. Properties of the mixture are calculated as follows:(14)$$P=P_{1}C+P_{2}\left(1-C\right)$$
where *P*—properties of the mixture (density, melting point, specific heat capacity, thermal diffusivity, viscosity, etc.), *P*_1_, *P*_2_—properties of the first and second materials at the actual calculated temperature, and *C*—volume fraction of the first material.

The second block of the LBM algorithm is needed to define the interaction in different interfaces (liquid–gas, solid–liquid, and solid–liquid–gas), mainly used for fluid flow (surface tension, wettability) but also for heat transfer (additionally, solid-gas interface).

The third block calculates macroscopic variables for fluid flow: density (1) and velocity (from (2) and (1)). Macroscopic velocity is used for the calculation of the equilibrium distribution function not only for the fluid flow problem (3), but also for the heat flow (7) and diffusion (11) problems. The absence of velocity in (7) and (11) corresponds to the diffusion equation (Fourier), while the presence of velocity corresponds to the diffusion–advection equation.

Collision operations are performed in the third block for fluid flow (3) and (4) and in the fourth block for heat flow (7) and (8) and diffusion (11) and (12).

The fifth block performs a streaming operation for all three problems according to (5), (9), and (13). The components of the distribution functions are transferred to the appropriate neighboring nodes.

The last block gives the boundary condition and is responsible for the movement of interfaces connected with fluid flow (liquid–gas, solid–liquid–gas). The movements of interfaces connected with melting and solidification (solid–liquid–gas and solid–liquid) are considered in the first block. Details of the different interfaces used in the model can be found in [20].

A key parameter for proper simulation and agreement on kinetics of different processes is the relaxation time for three problems. They are calculated basing on simulation parameters (time step, lattice length, velocity model D3Q19) and material properties (kinematic viscosity *ν*, thermal diffusivity *α*, diffusion coefficient *D*) as follows:(15)$$\tau_{f}=\frac{3\nu \Delta t}{\Delta x^{2}}+0.5$$
(16)$$\tau_{h}=\frac{3\alpha \Delta t}{\Delta x^{2}}+0.5$$
(17)$$\tau_{d}=\frac{3D\Delta t}{\Delta x^{2}}+0.5$$

## 3. Simulation and Results

This stage of the research involves the development and verification of the model of the PBF process objected to the fabrication of grade materials. For the verification of the model, the data published in [22,23] were used. It explains the modeling scheme and some parameters of material distribution.

### 3.1. Materils

AISI 316L austenitic steel (material 1) and 18Ni maraging 300 martensitic steel (material 2) were chosen for the simulations. The physical properties of these materials are collected in Table 1. The comparison of the diffusion coefficient with the kinematic viscosity and thermal diffusivity shows the diffusion coefficient is several orders of magnitude smaller and could be neglected, and only advection and mixing can be taken into account. Since LBM considers the diffusion–advection equation, it allows us to model almost pure advection Taking into account the velocity in the calculation of the equilibrium distribution, Function (11) corresponds to advection (and mixing), while the relaxation time in the determination of the output distribution Function (12) and (17) determines the diffusion rate. Both materials are closely related to the movement (advection) of the fluid. When two streams with different fractions arrive at the same node (cell), the volume contents of both materials are summed to obtain the resulting fraction. When there is movement of the material in the liquid, the fraction can be changed. When the liquid is motionless, only diffusion is present.

### 3.2. Simulation of Powder Filling and Deposition

Figure 3 presents the simulation scheme and two examples of filling the powder chamber with a wall rotation angle equal to 0 and 25°. The simulations were performed for the following cases.

Three rotation angles: (a) 0°, (b) 25°, and (c) 45°.Size of the particles: (a) equal to 40 μm, (b) material 1–35 μm, material 2–45 μm, (c) Gaussian distribution, material 1 average size 35 μm, material 2–45 μm, dispersion for both materials—5 μm.Friction coefficient: (a) 0, (b) 0.2, (c) 0.6.

The deposition process from the powder chamber to building platform with the coater is shown in Figure 4. The process is shown for a wall angle equal to 0° and ten layers with a friction coefficient of 0.2 and particles of 40 μm.

The deposition results for the case of different rotation angles, Gaussian distribution of the particle size, and coefficient of friction of 0.2 are shown in Figure 5. The white figures in Figure 5b represent an example of domains, which are transferred to the LBM model for further simulations. The location and sizes of the transferred domains are the same for all simulation cases.

### 3.3. Simulation of the PBF Process with LBM

The powder deposition results are transferred from the Unity-based model to the LBM model. In fact, the simulations in Unity and LBM are performed on different scales and different domains (Figure 5b). As mentioned above, the scale of the Unity-based model is 20:1 to respond to real sizes. LBM operates with dimensionless units of length and times. The lattice size in the LBM model, that is, the distance between the neighboring nodes, is equal to 1.25 μm. The entire domain is not simulated; only single tracks at different locations are. This allows us to reduce the simulation time.

Various simulations of the PBF process were performed, mainly at the angle of 0 and 25°, and the results of the distribution of the materials were compared with the distribution of the particles before the process. The simulation tracks are located on the axis of the building platform (1250 × 1250 μm) and at a distance of 375 μm from the axis symmetrically on both sides, as shown in Figure 5b. The size of the LBM modeling space is 960 × 160 × 80 μm or 768 × 128 × 64 nodes. The laser power is 300 W, the scan speed is 1000 mm/s, the laser movement distance is 850 μm (680 lattice lengths), and the laser spot diameter is 80 μm (64 lattice lengths).

The results of one of the simulation cases of the PBF process are presented in Figure 6, Figure 7 and Figure 8. Figure 6 presents a process as the 3D perspective view with a rendering at five moments of time. The left snapshot presents the initial particle deposition before the PBF process transferred from the Unity-based model. The next three snapshots present three moments of time during the processing of one track—at the beginning, middle, and end of the track. The last snapshot demonstrates the track after the whole material is solidified. Unprocessed particles are shown in grades of red and blue. Solidified material is shown in dark colors, with the fractions of red and blue corresponding to the fractions of the first and second materials. Materials in the liquid state are shown in color as solidified materials but with the addition of green grade. The laser beam is shown in green.

Other results are presented in Figure 7 and Figure 8. They show the materials before and after the PBF process as a longitudinal cross section on the track axis and three transverse cross sections located at the center of the modeled space and at the distance of 300 μm on both sides of the center. Unprocessed particles on the longitudinal cross sections (Figure 7) are shown in lighter colors; solidified materials are shown in deeper colors (red and deep blue); and a mixture is shown in an appropriate fraction of red and blue. Overheated molten material can melt some particles completely or partially and flow beneath them, filling the pores between unmolten particles. The flowing material initiates mixing, which also involves diffusion. However, for the single track on the microscale, in which the simulations with LBM were performed, highly mixed areas occupy only a small fraction, especially when cross sections are considered. Near the surface of the melting pool, the mixing is much higher than near its bottom. The higher mixing can be seen in the 3D view (Figure 6). This can be explained by the higher flow velocity and the additional fast stirring along with the slow diffusion. Near the bottom, the temperature is lower, the viscosity is lower, and the processes of stirring and diffusion are slower; the modeling results reflect this. Unprocessed and solidified materials are shown in the same colors on the transverse cross sections (Figure 8).

### 3.4. Analisys of Material Distribution

Some deposition results as a material distribution are presented in Figure 9. The left column represents the distribution before simulation of the PBF process. The volume fraction of the materials is in the range of 55–62%; the rest is occupied with gas. After the process, there is no gas and the total fraction of materials reaches 100%. The results before and after the process are almost the same; the error can be treated as a small random error. A transient zone from one material to another spreads from 0.25 mm for the angle of 0° through approximately 0.4–0.5 mm for the angle of 25° to the entire modeling space for the angle of 45°. The distribution is almost the same for different locations of the tracks for the rotation angle of 0° (Figure 9a); it can also be seen in Figure 4 and Figure 5a. The increase in the rotation angle (Figure 9a,c) allows the transient zone to spread, but the difference in its location and length become uncontrolled. The heterogeneity of the distribution can also be seen in Figure 5b,c. There are several factors that influence the results obtained in the simulations. Even if the other conditions are the same, it depends on which powder is scraped first with the coater blade. In addition, smaller particles are deposited first; this is especially important when the average size of powder particles of different materials is different. The higher density particles are deposed first in one layer and go deeper when they are deposed in several layers without melting, but in the case of the materials presented in this paper, the density of the materials is almost the same, and this factor does not play a notable role. The influence of the friction coefficient on the deposition is also noticed, but it is difficult to formulate.

It is very difficult to perform a quantitative comparison with experimental data. The experimental results presented in [23] do not allow for the reproduction and repetition of the study in many details. That is why the comparison can be treated as qualitative only. The simulation can be considered qualitatively consistent with the results of the experimental studies presented in [23] and Figure 10. The experimental results show that diffusion can be neglected and either element can be chosen to define the fraction of the materials, while material fraction in simulations can be obtained directly [23]. The angle of the wall assumes the length of the transient gradient zone of 20 mm; perhaps the angle in the experimental studies was of 15–20°. The length of the powder chamber and the building platform was approximately 60–80 mm. The specimen sizes were 30 × 10 mm. Pairs of specimens were located symmetrically on a building platform at a distance of approximately 30 mm from each other. Two pairs differed in the order of the powders in the powder chamber. Many other parameters are unclear.

The results in [23] demonstrate significant differences in the distribution of different elements that define different materials. The distributions are different due to the different locations of the specimens and the order of the powder in the powder chamber. The lack of information does not allow the author to explain the differences obtained in experimental studies, but the studies presented in this paper can assume that these differences can be explained by the location of the specimens and the different sizes of the particles.

## 4. Discussion

The results presented in the previous sections show the possibility of modeling a complex PBF process with the mixture of two materials to obtain graded materials. Analysis of the distribution of the materials in the final mixture and its comparison with experimental data allows us to conclude that the complex Unity–LBM model gives adequate results, at least qualitatively similar to the real PBF process. The analysis was performed not to show the effectiveness of the FGM fabrication method, but to demonstrate that the LBM-based model can be applied to the simulation of the fabrication of functionally graded material in additive manufacturing. The Unity-based model is of its own independent importance. Simulations by the Unity-based model of the deposition of materials with different physical and mechanical properties allow for the simulation, study, modeling, and optimization of deposition schemes and methods for graded materials because the PBF process that provides homogenization does not significantly influence the final material distribution. Future research may be connected with further verification and validation of the model, modeling different materials, and the optimization of the methods of manufacturing functionally graded materials in the PFD processes.

## 5. Conclusions

This paper presents the results of the further development of the platform for modeling multi-material multilayer PBF processes directed at the simulation of the fabrication of functionally graded materials. To achieve this goal, Unity- and LBM-based models were modified. The Unity-based model allows for the simulation of the deposition of materials with different mechanical properties to create input information for the LBM-based model. The LBM-based model was extended with several new elements. The diffusion phenomenon with material mixing was added; it led to the development and implementation of the diffusion model, which influenced the introduction of the third set of variables and distribution functions. In addition to diffusion, changes were implemented in the fluid and heat flow models. The fraction of the materials was introduced, which defines the physical, mechanical, and thermal properties of the mixture and the effect on the simulation parameters. The comparison of the results obtained in the simulation with the experimental data shows that they are consistent. The presented model can be used for the simulation, study, modeling, and optimization of the production of functionally graded materials in PBF processes.

## Figures and Tables

**Figure 1 entropy-27-00020-f001:**
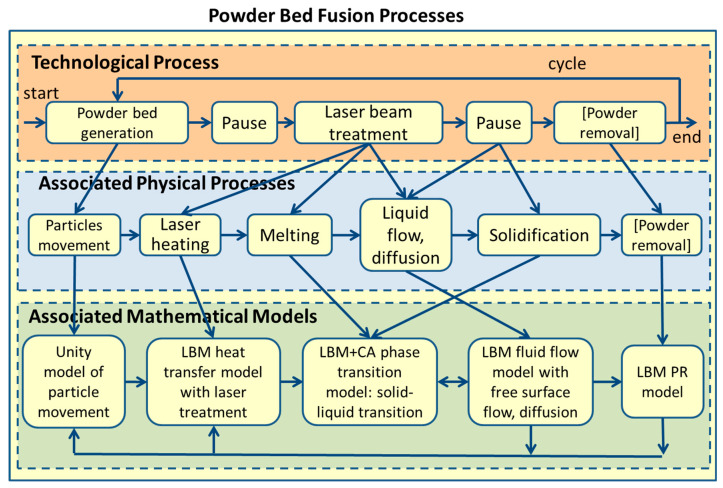
The PBF process with associated physical phenomena and extended submodels for modeling of mixture of two materials and diffusion.

**Figure 2 entropy-27-00020-f002:**
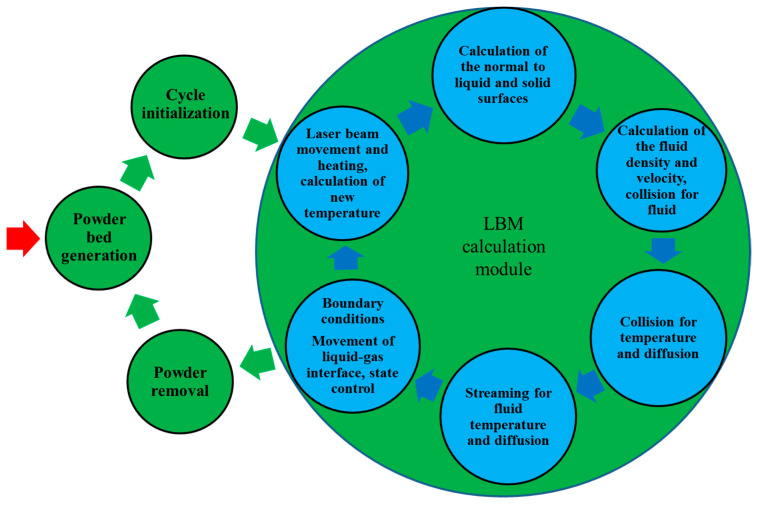
Graphical representation of the main algorithm.

**Figure 3 entropy-27-00020-f003:**
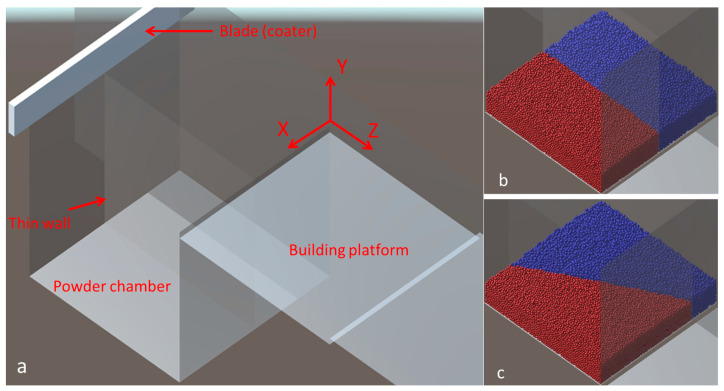
Modeling scene for simulation of the deposition of the powder of two materials shown in different colors (**a**) and two cases of filling of the powder chamber (**b**,**c**).

**Figure 4 entropy-27-00020-f004:**
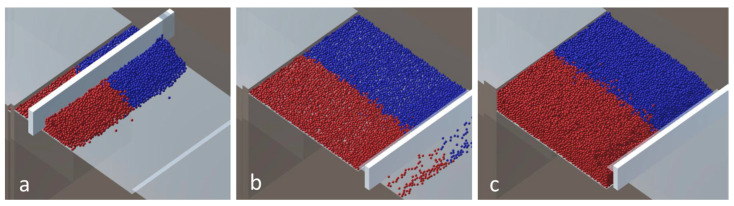
Powder deposition process (**a**) beginning of the first layer, (**b**) the end of the first layer, (**c**) results with 10 layers. Different colors presents different materials.

**Figure 5 entropy-27-00020-f005:**
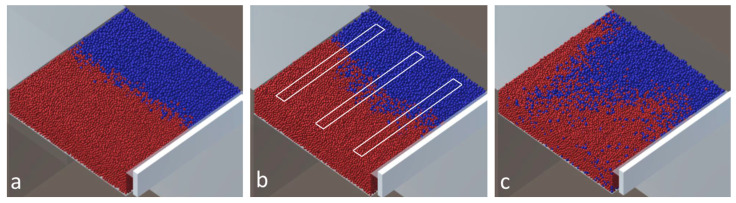
Particle powder deposition with Gaussian size distribution and different wall rotation angle: (**a**) 0°, (**b**) 25°, (**c**) 45°. Different colors present different materials.

**Figure 6 entropy-27-00020-f006:**
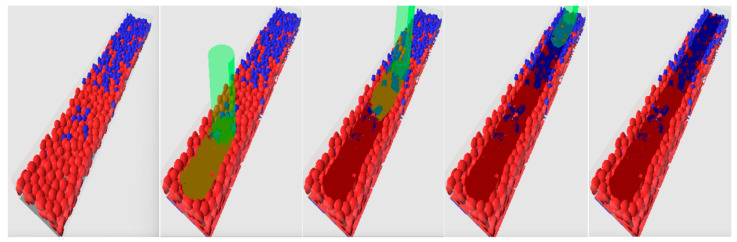
3D view of several stages of the simulated PBF process with LBM. Colors are explained in the text.

**Figure 7 entropy-27-00020-f007:**
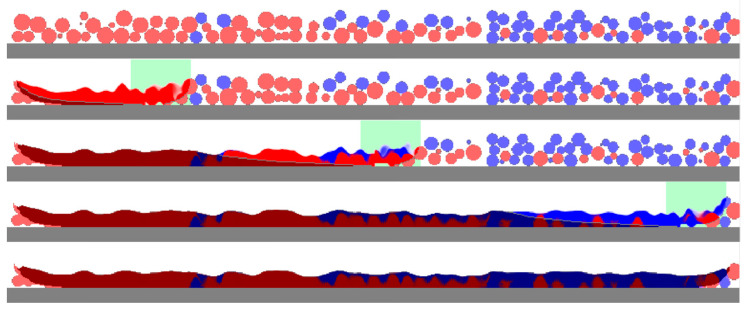
A longitudinal cross section of the particle of two materials before, during, and after the PBF process.

**Figure 8 entropy-27-00020-f008:**
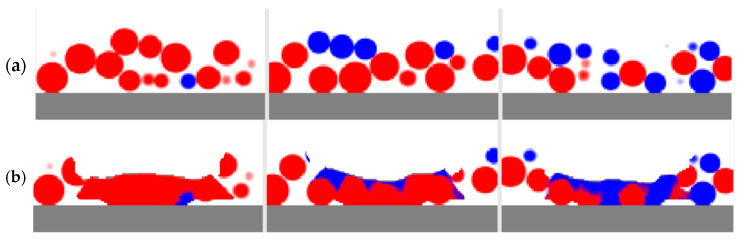
Transverse cross sections in three locations before (**a**) and after (**b**) the PBF process simulated with LBM.

**Figure 9 entropy-27-00020-f009:**
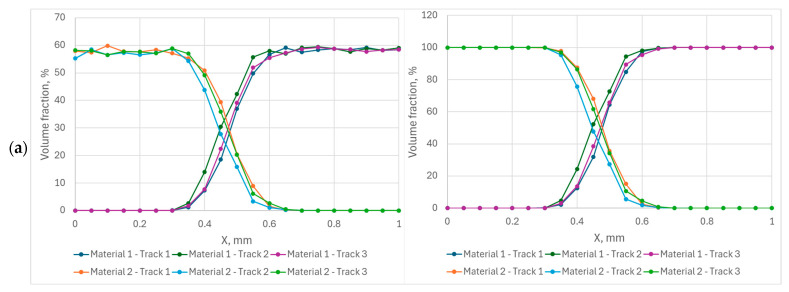
Distribution of two materials before (**left column**) and after (**right column**) the simulated PBF process for different rotation angles of the wall: (**a**) 0°, (**b**) 25° (**c**) 45°.

**Figure 10 entropy-27-00020-f010:**
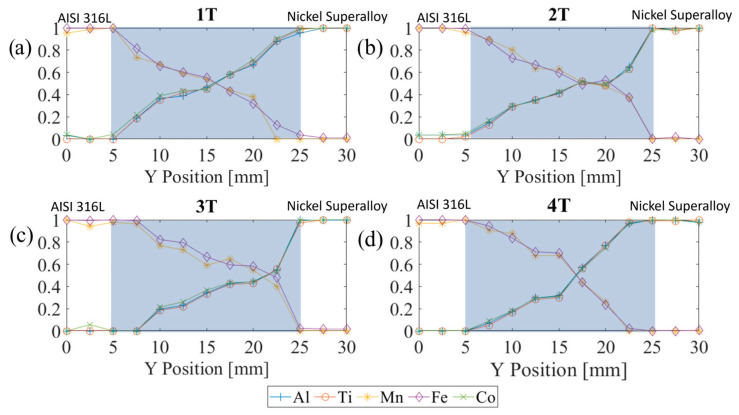
Results of chemical analysis performed on samples from different locations in [23]. Different orientation of the dividing wall (**a**,**b**)—left orientation, (**c**,**d**)—right orientation. AiSI 316L closer to the building platform (**b**,**c**), Ni superalloy closer to the platform (**a**,**d**).

**Table 1 entropy-27-00020-t001:** Physical properties of AISI 316L austenitic steel and 18Ni maraging 300 martensitic steel.

Properties	AISI 316L	18Ni Maraging
Density, kg/m^3^	8000	8100
Melting point, K	1648–1673	1686
Specific heat capacity, J/(kg·K)	500	452
Thermal conductivity, W/(m K)	14.0–15.9	25.5
Thermal diffusivity, m^2^/s	3.75 × 10^−6^	5.5 × 10^−6^
Convective heat transfer coefficient, W/(m^2^ K)	50	50
Coefficient of thermal expansion, K^−1^	16–18 × 10^−6^	11.3 × 10^−6^
Dynamic viscosity, kg/(m s)	2.4–3.3 × 10^−3^	
Kinematic viscosity, m^2^/s	0.3–0.41 × 10^−6^	
Surface tension, N/m × 10^−4^ [30]	1.86–1.54 (*T*-1811.15)	
Diffusion coefficient, m^2^/s	5 × 10^−9^–10^−7^.	
Relaxation times for liquid flow (15)	0.577	
Relaxation times for heat flow (16)	1.22	
Relaxation times for diffusion (17)	0.505	

## Data Availability

The original contributions presented in this study are included in the article material. Further inquiries can be directed to the corresponding author.

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
