# Peer review of "Lattice Boltzmann Modeling of Additive Manufacturing of Functionally Graded Materials"

_entropy, 2024, doi:10.3390/e27010020_

Round 1
Reviewer 1 Report
Comments and Suggestions for Authors
This work presents an extension of multi physics numerical solver, previously presented in Refs[19-21], for the study of Powder Bed Fusion. I am quite impressed by the framework developed by the authors, and the topic is surely interesting and suitable for publication on ''Entropy''.
Unfortunately, however, I cannot recommend this work for publication, at least not in its present form. In my opinion there are currently two main issues:
1) I find the description of the method very hard to follow and understand. Given the complexity of the framework I find justified to refer the reader to previous publications for details, but at least the main (macroscopic) equations that are being solved should be given explicitly. If I understand, the only novel bit added is in section 2.3.3, where a passive scalar tracks the fraction of material in the mixture. What is \tau_d and how is its value selected? In the conclusions is seems that this part requires modifications also to fluid and heat flow models, but there is no hint of this in 2.3.1 and 2.3.2.
Moreover, I am not sure what the author means with "unity" model, are we talking of the game engine? I could not find any description of the unity model in the references provided. Comparing fig 1 with the one provided in 0.1007/s00170-023-12138-x, I see that the particle movements was simulated with a cellular-automata scheme, this seems to have been replaced in the present work with the aforementioned unity model. Please clarify (and maybe compare/validate).
2) The results presented are fully qualitative, and the discussion very shallow. Fig.6 is virtually indistinguishable from those previously presented in 10.1016/j.simpat.2024.103009, where diffusion was neglected. To start, is it possible to have a more quantitative comparison with previous results? Is it possible to assess if and when diffusion play a role here (e.g., in which parameter range)? Similar comment for fig 8 (please also consider a different choice of colors for pre-post PBF since they are virtually impossible to distinguish). The simplification that all materials diffuse at the same rate might be reasonable of very crude depending on the specific study case, please comment on that. Finally, the comparison in Sec.3.4 with experimental results is potentially very interesting, although hard since many parameters are unknown. I find very hard to compare Fig9 and Fig10, it might be helpful to put the relevant panels side-by-side.
Reviewer 2 Report
Comments and Suggestions for Authors
The manuscript presents a novel approach to simulating the powder bed fusion (PBF) process for the fabrication of functionally graded materials (FGMs) using a combination of Unity-based deposition and Lattice Boltzmann method (LBM)-based process models. The study introduces a diffusion model that allows the simulation of material mixtures, specifically AISI 316L austenitic steel and 18Ni maraging 300 martensitic steel. The research is timely and relevant given the increasing interest in additive manufacturing and the development of FGMs for various applications.
1. What is the colors in Fig. 6 and Fig.8 strands for?
2. How does the mixture flows in LBM? In streaming and collision operations, how to consider the mixture?
